# 🐋 BALEEN: SELF-INTERPRETABLE, ROBUST SSMS WITH STOCHASTIC SELECTIVE MEMORY

## ABSTRACT

We introduce *Baleen*, a family of state space models that unifies **stochastic selection** with **information bottleneck** to build interpretable and robust long-context learners. Unlike Mamba/Mamba2's deterministic gates, Baleen treats selection as a random variable and regularizes it with a closed-form KL to a sparsity prior: (i) *Baleen-B* samples Bernoulli state-transition gates; (ii) *Baleen-E* samples Exponential time-intervals. This yields an explicit trade-off between retention and compression and exposes token-level selection heatmaps at inference for self-interpretation. On language benchmarks, Baleen improves average accuracy over Mamba2 by +0.95 at 370M pretraining and +1.38 at 7B finetuning. Baleen delivers stronger robustness to localized perturbations and adversarial attacks: under CIFAR-10 sequence perturbation, prefix damage falls to 0.6% vs 26.5% for Mamba2 (average under attacks 0.542 vs 0.385). Finally, Baleen's self-interpretations outperform IG/Grad-CAM on average fidelity across four text classification tasks. We will release our Baleen-7B models on Hugging Face with code, checkpoints, and an interactive selection-heatmap demo.

## 1 INTRODUCTION

As competitive successors to transformers (Vaswani et al., 2017), State Space Models (SSMs) have emerged as a powerful neural architecture for sequence-to-sequence modeling, demonstrating impressive efficiency in processing long sequential data (Gu et al., 2021a; Gu & Dao, 2023; Dao & Gu, 2024). SSMs are derived from discretizing a linear time-invariant dynamical system (Gu et al., 2020; 2021b; 2022b). Each step $t$ corresponds to a token and involves two key components: *i)* a fixed-dimension hidden *memory state* matrix $\boldsymbol{H}_{t-1}$, which maintains a running summarization of past token embeddings $\boldsymbol{X}_{1:t-1}$ for prediction, and *ii)* a parameterized recurrence, which applies a *state transition* matrix $\boldsymbol{A}_t$ to transform memory states while sequentially integrating per-token representations into the memory. Recent innovations in SSMs sought to refine the recurrence mechanism to effectively compress context of varying length into a fixed-size memory (Yang et al., 2023; 2024b; Sun et al., 2023; Liu et al., 2024; Sun et al., 2024).

Linear time-variant SSMs (Gu et al., 2021a) have exhibited less satisfactory performance in natural language modeling. The seminal work *Mamba* (Gu & Dao, 2023) introduces a selection mechanism that conditions recurrence parameters on the input sequence with non-linearity. This selection mechanism is designed to enable more flexible context filtering and dynamic selection of important tokens by leveraging the state transition matrix as an input-adaptive gating function, where the parametrization of $\boldsymbol{A}_t$ involves the input token embedding $\boldsymbol{X}_t$.

However, since Mamba is trained solely to minimize the prediction error at the population level, there is no explicit constraint enforcing it to retain only the useful context in the memory state. As a result, the memory may capture spuriously correlated information from the context, reducing its capacity to accommodate new, relevant information during recurrence, and deteriorating vulnerability to noisy perturbations (Wang et al., 2024; Poli et al., 2024; Park et al., 2024). Meanwhile, although Mamba is intended to be selective, extracting interpretable patterns to indicate what tokens matter remains challenging. Recent works (Ali et al., 2024; Jafari et al., 2024) attempt to explain Mamba by recovering token-wise importance weights. However, it seems these methods struggle to provide reliable interpretation in language tasks, as Mamba overwhelmingly prioritizes local tokens (Wang et al., 2024).

To address these limitations, we introduce *Baleen*, a family of information-bottleneck SSM architectures that explicitly optimizes for the context compression rate through an stochastic selection mechanism, thereby enabling better generalization, robustness to noisy perturbations, and intrinsic interpretability via stochastic selection gates. Grounded in the information bottleneck principle (Tishby et al., 2000), we conceptualize the hidden memory states in SSMs as an information bottleneck, requiring them to achieve a minimal representation of past context while retaining sufficient information for accurate predictions. From this formulation, we derive two novel SSM architectures equipped with stochastic selection mechanisms and an associated training objective, instantiated through two alternative strategies for randomness modeling.

- **Principled framework of IB-gated SSM with stochastic selection:** We introduce **Baleen** model family, based on a novel state space information bottleneck formulation that converts Mamba's deterministic selection into stochastic selection with a tractable variational objective. We implement two efficient, plug-in variants: Bernoulli gating of state-transition entries and Exponential timestep sampling, each with a closed-form KL regularizer, preserving the linear-time Mamba kernel.

- **Self-interpretation by design, not post-hoc:** Treating selection as a random variable yields token-level selection heatmaps from expected gates, requiring no extra forward/backward passes and enabling faithful top-$k$ rationales. We also provide an information-theoretic critique of Mamba: its MLE objective maximizes $\mathbb{I}(\boldsymbol{Y}; \boldsymbol{H})$ without penalizing $\mathbb{I}(\boldsymbol{H}; \boldsymbol{X})$, while Baleen explicitly trades off these terms.

- **Strong prediction accuracy, higher fidelity of interpretation, and robustness to adversarial attack:** across 11 language tasks, Baleen outperforms Mamba2 at both 370M (+0.95 avg ) and 7B (+1.38 avg) and surpass all other baselines. Baleen's self-interpretations surpass IG/Grad-CAM/Grad×Input in average fidelity on SNLI/IMDb/SST2/RT. Under pixel-sequence perturbations and adversarial prompt attacks ranging from the character level to the sentence level, Baleen demonstrates markedly greater robustness (e.g., prefix [0:32] drop 0.57% vs 26.51% for Mamba2; attack-averaged accuracy 0.542 vs 0.385 on CIFAR-10).

## 2 PRELIMINARIES

**Mamba.** In this work, we focus on discrete-time SSMs with real-valued diagonal state transition matrices and zero-order hold discretization rule, a design that underpins the recent success of SSM-based LLMs(Gupta et al., 2022; Gu et al., 2022a; Gu & Dao, 2023).

Let $\boldsymbol{X} := [\boldsymbol{x}_1^\top, \cdots, \boldsymbol{x}_T^\top]^\top \in \mathbb{R}^{T \times D}$ be an input sequence, where $T$ is the sequence length and $D$ is the token embedding dimension. A general SSM layer introduces a group of parameters $\{(\boldsymbol{A}_t, \boldsymbol{B}_t, \boldsymbol{C}_t, \boldsymbol{\Delta}_t)\}_{t \in [T]}$ to process the sequence according to the following equations:

$$\boldsymbol{H}_t = \boldsymbol{A}_t \odot \boldsymbol{H}_{t-1} + \boldsymbol{B}_t \odot (\mathbf{1}_N \boldsymbol{x}_t^\top), \quad \widehat{\boldsymbol{y}}_t = \boldsymbol{C}_t^\top \boldsymbol{H}_t \tag{1}$$

where $\odot$ denotes element-wise multiplication, $\boldsymbol{A}_t = \exp(\mathring{\boldsymbol{A}}_t \operatorname{diag}(\boldsymbol{\Delta}_t))$, $\boldsymbol{B}_t = \mathring{\boldsymbol{B}}_t \boldsymbol{\Delta}_t^\top$, $\boldsymbol{\Delta}_t \in \mathbb{R}_+^D$, $\boldsymbol{A}_t \in \mathbb{R}_-^{N \times D}$ $\boldsymbol{B}_t \in \mathbb{R}^N$, $\boldsymbol{C}_t \in \mathbb{R}^{N \times D}$ for all $t \in [T]$. Note that $\boldsymbol{\Delta}_t$ is strictly positive plus $\mathring{\boldsymbol{A}}_t$ is strictly negative, ensuring $\boldsymbol{A}_t \in (0, 1)^{N \times D}$. Define $\boldsymbol{H} := [\boldsymbol{H}_t \in \mathbb{R}^{N \times D}]_{t \in [T]}$ as a $T \times N \times D$ tensor to represent the intermediate *memory states*, and $\widehat{\boldsymbol{Y}} := [\widehat{\boldsymbol{y}}_1^\top, \cdots, \widehat{\boldsymbol{y}}_T^\top]^\top \in \mathbb{R}^{T \times D}$ denotes the output sequence. Next, the operation $\boldsymbol{B}_t \odot (\mathbf{1}_N \boldsymbol{x}_t^\top)$ encodes the input tokens into the hidden state space. And finally, $\boldsymbol{C}_t^\top \boldsymbol{H}_t$ decodes the memory state to generate the prediction $\widehat{\boldsymbol{y}}_t$ for the $t$-th token.

In *S4* (Gu et al., 2021a), the parameters $\{(\mathring{\boldsymbol{A}}_t, \mathring{\boldsymbol{B}}_t, \boldsymbol{C}_t, \boldsymbol{\Delta}_t)\}_{t \in [T]}$ are directly learned and remain constant across different token positions. While this approach is effective for certain long-sequence tasks (Gu et al., 2020; 2022b), the time-invariant linearity limits its ability to capture more complex and dynamic signals within the context. Subsequently, *Mamba* (Gu & Dao, 2023) conditions the parameters on the input sequence itself. To be specific, Mamba takes the following form of parameterization:

$$\mathring{\boldsymbol{A}}_t = \boldsymbol{A}, \quad \mathring{\boldsymbol{B}}_t = \boldsymbol{W}_B \boldsymbol{x}_t, \quad \boldsymbol{C}_t = (\boldsymbol{W}_C \boldsymbol{x}_t) \mathbf{1}_D^\top, \quad \boldsymbol{\Delta}_t = \operatorname{softplus}(\boldsymbol{W}_\Delta \boldsymbol{x}_t), \tag{2a}$$

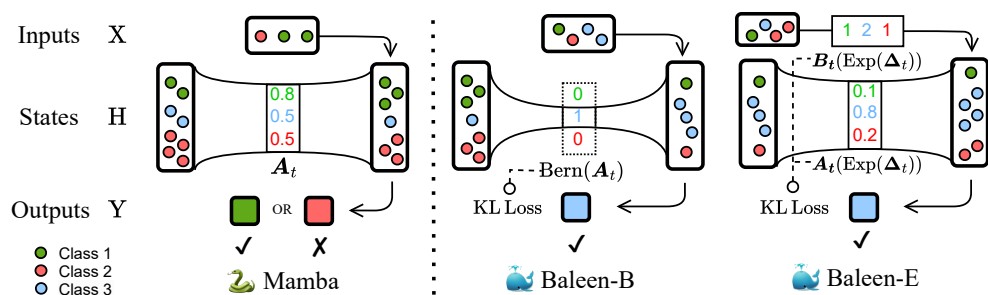

Figure 1: Overview of Baleen compared with previous SSMs such as Mamba. The information bottleneck injects random noise and enhances the selection of the state transition matrix by adapting the deterministic $\boldsymbol{A}_t$ or $\boldsymbol{\Delta}_t$ to the randomly sampled variable. For simplicity, we omit deterministic $\boldsymbol{B}_t$ and $\boldsymbol{C}_t$.

where the learnable parameters in an SSM layer includes $\boldsymbol{A} \in \mathbb{R}^{N \times D}_{-}$ being invariant across tokens, $\boldsymbol{W}_B \in \mathbb{R}^{N \times D}$, $\boldsymbol{W}_C \in \mathbb{R}^{N \times D}$, and $\boldsymbol{W}_\Delta \in \mathbb{R}^{D \times D}$. In Mamba2, Dao & Gu (2024) proposes to impose additional structures on the parameter space. Specifically, $\boldsymbol{A}$ is simplified to $\boldsymbol{1}_N \boldsymbol{\alpha}^\top$ for some $\boldsymbol{\alpha} \in \mathbb{R}^D$ and $\boldsymbol{W}_\Delta$ becomes a row-block matrix.

**Information Bottleneck.** The information bottleneck principle (Tishby et al., 2000) offers a principled approach to balancing the trade-off between representation compactness and predictive capability. It relies on the concept of minimal sufficient statistics to encode information about a target variable $Y \in \mathcal{Y}$ into a representation $Z = \Phi(X) \in \mathcal{Z}$ derived from input $X \in \mathcal{X}$. The principle enforces a regularization on $Z$ by minimizing the mutual information $I(X; Z)$ between the input and the representation, while maximizing the mutual information $I(Y; Z)$ between the representation and the target. Formally, the objective is given by:

$$\min_{\Theta, \Phi} -\mathbb{I}(\boldsymbol{Y}; \boldsymbol{Z}) + \beta \mathbb{I}(\boldsymbol{Z}; \boldsymbol{X}) \tag{3}$$

where $\beta$ is a hyperparameter to control the balance, $\boldsymbol{Z}$ follows conditional distribution $\mathbb{P}^\Phi(\boldsymbol{Z}|\boldsymbol{X})$ parametrized by an encoder $\Phi$ and $\boldsymbol{Y}$ follows conditional distribution $\mathbb{P}^\Theta(\boldsymbol{Y}|\boldsymbol{H})$ parametrized by a decoder $\Theta$.

Directly estimating the loss function in Eq. 3 can be computationally infeasible. Previous work Tishby et al. (2000); Alemi et al. (2016); Wu et al. (2020) have dereived a tractable variation upper bound (up to a constant) as below:

$$\min_{\Theta, \Phi} - \mathbb{E}\left[\log \mathbb{P}^\Theta(\boldsymbol{Y}|\boldsymbol{Z})\right] + \beta \mathbb{E}\left[D_{\mathrm{KL}}(\mathbb{P}^\Phi(\boldsymbol{Z}|\boldsymbol{X})\|\mathbb{Q}(\boldsymbol{Z}))\right] \tag{4}$$

## 3 BALEEN: STOCHASTIC SELECTION VIA INFORMATION BOTTLENECK

In this section, we present our approach to improving SSMs via information bottleneck. In Sec. 3.1, we introduce the state space information bottleneck framework, designed to enhance the context compression capability, generalization, and interpretability of SSMs. In Sec. 3.2, we propose two novel SSM architectures that incorporates principled randomness modeling within the SSIB framework. Finally, in Sec. 3.3, we demonstrate how our architectures is equipped with inherent interpretability for understanding the behavior of SSM-based LLMs.

### 3.1 STATE SPACE INFORMATION BOTTLENECK

We focus on sequence-to-sequence modeling tasks using the SSM architecture. Let $\mathcal{X}, \mathcal{Y} \subseteq \mathbb{R}^{T \times D}$ represent the domains of input and target sequences, respectively. The training set consists of IID samples from the unknown joint distribution $\mathbb{P}(\boldsymbol{X}, \boldsymbol{Y}) = \mathbb{P}(\boldsymbol{Y}|\boldsymbol{X})\mathbb{P}(\boldsymbol{X})$[1], which is supported

---

[1]When referring to a probability density or mass function (PDF or PMF), we will omit the subscripts specifying the random variables it is defined over, as long as they are clear from the context according to the PDF or PMF's arguments or parameters.

on $\mathcal{X}$ and $\mathcal{Y}$. In the case of causal language modeling (Radford et al., 2018; 2019; Brown et al., 2020), $\mathbb{P}(\boldsymbol{Y}|\boldsymbol{X}) = \delta_{\boldsymbol{X}^*}(\boldsymbol{Y})$, where $\boldsymbol{X}^* = [\boldsymbol{x}_2^\top, \cdots, \boldsymbol{x}_T^\top, \mathrm{EOS}]^\top$ is a shifted version of $\boldsymbol{X}$ with EOS (special token) padding.

We aim to train a sequence-to-sequence SSM following Eq. 1: $\mathrm{SSM}_{\Theta,\Phi} = g_\Theta \circ h_\Phi : \mathcal{X} \to \mathcal{Y}$, where *contextual encoder* $h_\Phi$ transforms subsequences of inputs $\boldsymbol{X}$ into the memory states $\boldsymbol{H}$ with parameters $\Phi = \{\boldsymbol{A}, \boldsymbol{W}_B, \boldsymbol{W}_\Delta\}$, and *predictor* $g_\Theta$ decodes the memory states $\boldsymbol{H}$ into the predicted sequence $\widehat{\boldsymbol{Y}} \in \mathcal{Y}$ with $\Theta = \{\boldsymbol{W}_C\}$ (see Eq. 1)[2]. An intrinsic property of SSMs is that $\boldsymbol{H}_t$ depends on $\boldsymbol{X}$ exclusively through its preceding tokens $\boldsymbol{X}_{\leq t} = [\boldsymbol{x}_1^\top, \cdots, \boldsymbol{x}_t^\top]^\top$. And the prediction $\widehat{\boldsymbol{y}}_t$ is conditionally independent of other variables given $\boldsymbol{H}_t$.

Naively applying the information bottleneck to SSMs presents unique challenges, as sequential data exhibit complex long-range dependencies that go beyond the IID assumptions (Alemi et al., 2016; Poole et al., 2019) or the local-dependence hypothesis (Wu et al., 2020; Miao et al., 2022). Based on the Markov property, a surrogate objective that upper-bounds the original objective can be derived as follows:

$$\min_{\Theta,\Phi} \underbrace{-\mathbb{E}_{\boldsymbol{H},\boldsymbol{Y}}\left[\sum_{t=1}^T \log \mathbb{P}^\Theta(\boldsymbol{y}_t|\boldsymbol{H}_t)\right]}_{L_{\mathrm{CE}}} + \beta\, \mathbb{E}_{\boldsymbol{X}}\underbrace{\left[\sum_{t=1}^T D_{\mathrm{KL}}(\mathbb{P}^\Phi(\boldsymbol{H}_t|\boldsymbol{H}_{t-1},\boldsymbol{X}_{\leq t})\|\mathbb{Q}(\boldsymbol{H}_t|\boldsymbol{H}_{t-1}))\right]}_{L_{\mathrm{KL}}} \quad (5)$$

Detailed derivations are deferred to Appendix A.2. We refer to the framework with learning objective in Eq. 5 as *State Space Information Bottleneck (SSIB)*.

**Information-Theoretic Pitfalls of Mamba.** Now we can revisit the training process of Mamba, revealing its equivalence to state space information bottleneck with $\beta = 0$, thus limited in compact selection. The training objective of Mamba is typically Maximum Likelihood Estimation (MLE): $\max_{\Theta,\Phi} \mathbb{E}_{\boldsymbol{X},\boldsymbol{Y}}[\sum_{t=1}^{T-1} \log \mathbb{P}^{\Theta,\Phi}(\boldsymbol{y}_t|\boldsymbol{X}_{\leq t})]$. We argue that simply maximizing the mutual information between $\boldsymbol{y}_t$ and $\boldsymbol{H}_t$ only ensures $\boldsymbol{H}_t$ retains sufficient information for predicting $\boldsymbol{y}_t$. However, this does not guarantee that $\boldsymbol{H}_t$ achieves the maximal compression of $\boldsymbol{X}_{\leq t}$ and only focuses on tokens useful for prediction, potentially leading to information loss as the context length grows larger. Moreover, $\boldsymbol{H}_t$ may capture *spurious correlations* between $\boldsymbol{X}_{\leq t}$ and $\boldsymbol{y}_t$, making the predictions highly susceptible to noise (Chen et al., 2018; Wang et al., 2024). Instead, our approach explicitly models the stochastic relationship between $\boldsymbol{X}$ and $\boldsymbol{H}$, enabling the denoising of spurious patterns in $\boldsymbol{H}$—a key factor for effective context compression, generalization, and interpretability.

### 3.2 BALEEN ARCHITECTURE AND LEARNING OBJECTIVE

In this subsection, we introduce the **Baleen** model family, which differs in *how random variables are modeled under the SSIB framework*. We begin by revisiting Eq. 1: the first term governs the transition of past memory, while the second term encodes the new token into the hidden memory state space. Since the state transition (i.e., $\boldsymbol{A}_t$) is the key component to randomize, whether randomness should also be incorporated into the new token encoding (i.e., $\boldsymbol{B}_t$) remains an open question. To investigate this, we first inject randomness solely into the state transition matrix, and then extend it to both the state transition and the new token encoding via the timestep variable $\boldsymbol{\Delta}_t$. The overall architectural design of SSIB is illustrated in Fig. 1.

**Bernoulli-Distributed Transition.** As mentioned before, each element in the memory state $\boldsymbol{H}_t$ represents a distinct component of past context (Gu et al., 2020), where a state transition value of one or zero determines whether the component is kept or removed from memory. It is naturally to model $\boldsymbol{A}_t$ as a variable follows Bernoulli distribution. In this case, $\boldsymbol{H}_t$ is an affine transformation of $\boldsymbol{A}_t$ and also follows a Bernoulli distribution. Moreover, each outcome of $\boldsymbol{H}_t$ corresponds to a unique outcome of $\boldsymbol{A}_t$ with the same probability. Therefore, the KL divergence is preserved under this affine transformation, i.e., the following equation holds:

$$D_{\mathrm{KL}}(\mathbb{P}(\boldsymbol{H}_t|\boldsymbol{H}_{t-1},\boldsymbol{X}_{\leq t})\|\mathbb{Q}(\boldsymbol{H}|\boldsymbol{H}_{t-1})) = D_{\mathrm{KL}}(\mathbb{P}(\boldsymbol{A}_t)\|\mathbb{Q}(\boldsymbol{A}_t)) \quad (6)$$

---

[2]Without loss of generality, we only consider SSMs with a single layer, without channel-mixing layer (Fu et al., 2022). We omit parameters for the embedding layer and the output head.

where $\mathbb{Q}(\boldsymbol{H}|\boldsymbol{H}_{t-1}))$ is the corresponding affine transformation of $\mathbb{Q}(\boldsymbol{A}_t)$.

Because variational bound Eq. 5 holds true for any prior distribution $\mathbb{Q}$, we can simply define the prior distribution of $\boldsymbol{A}_t$ as $\text{Bern}(p_t \mathbf{1}_N \mathbf{1}_D^\top)$ where $p_t \in [0, 1]$ is a hyper-parameter. In training process, we implement the sampling process using the gumbel-softmax technique to ensure differentiability (Bengio et al., 2013; Jang et al., 2016; Maddison et al., 2016). Then the KL loss in Eq. 5 is simplified as follows:

$$L_{\text{KL}} = \mathbb{E}\left[\sum_{t,n,d} \boldsymbol{A}_t^{(n,d)} \log \frac{\boldsymbol{A}_t^{(n,d)}}{p_t} + (1 - \boldsymbol{A}_t^{(n,d)}) \log \frac{1 - \boldsymbol{A}_t^{(n,d)}}{1 - p_t}\right] \tag{7}$$

where the summation is taken over $t \in [T], n \in [N], d \in [D]$. Detailed derivations can be found in Appendix A.3.

**Exponential-Distributed Timestep.** To simultaneously inject randomness into both $\boldsymbol{A}_t$ and $\boldsymbol{B}_t$ is challenging, especially $\boldsymbol{A}$ and $\boldsymbol{B}_t$ should conform to different distributions even discrete and continous are not the same. Based on our observation on both $\boldsymbol{A}$ and $\boldsymbol{B}_t$ are parameterized with $\boldsymbol{\Delta}_t$, we may assume $\boldsymbol{\Delta}_t$ a random variable to implicitly affect both terms. In SSMs, $\boldsymbol{\Delta}_t$ is often interpreted as the discretization timestep from the perspective of a dynamic system (Gu & Dao, 2023), while the exponential distribution is classically used to model the time interval. Motivated by this connection, we propose to redefine $\boldsymbol{\Delta}_t$ in Eq. 2a as a random variable following Exponential distribution. In this case, $\boldsymbol{H}_t$ has an explicit expression form:

$$\boldsymbol{H}_t = \boldsymbol{H}_{t-1} \exp(\mathring{\boldsymbol{A}}_t \operatorname{diag}(\boldsymbol{\Delta}_t)) + \mathbf{1}_N \boldsymbol{x}_t \mathring{\boldsymbol{B}}_t \boldsymbol{\Delta}_t \tag{8}$$

where the first term follows Pareto distribution and the second term still follows Exponentil distribution. their sum does not belong to any well-known class of distributions. Since $\boldsymbol{H}_t$ as a function of $\boldsymbol{\Delta}_t$ is not invertible, the KL divergence in Eq. 5 cannot be directly reduced to that between $\boldsymbol{\Delta}_t$ and its prior like Eq. 6 shows. Nevertheless, we can show that this KL divergence serves as an upper bound of the original one, which allows us to directly optimize with respect to $\boldsymbol{\Delta}_t$. Further details are provided in Appendix A.3.

We then choose the prior distribution $\mathbb{Q}(\boldsymbol{\Delta}_t)$ as an Exponential distribution, $\text{Exp}(\lambda_t \mathbf{1}_D)$, which yields an upper bound on the KL divergence in Eq. 5, as shown below:

$$L_{\text{KL}} = \mathbb{E}\left[\sum_{t,d} \log \frac{\boldsymbol{\Delta}_t^{(d)}}{\lambda_t} - (\boldsymbol{\Delta}_t^{(d)} - \lambda_t)\frac{1}{\boldsymbol{\Delta}_t^{(d)}}\right], \tag{9}$$

where the summation is taken over $t \in [T], d \in [D]$.

We name the two novel architectures with stochastic selection *Baleen-B* (Bernoulli) and *Baleen-E* (Exponential). While Baleen-B polarizes the state transition values to ensure a compact representation, Baleen-E implements a *trade-off* between incorporating more past memory and more new encoded token information. This behavior can be explained as follows: when $\boldsymbol{\Delta}_t$ tends to infinity, the first term in Eq. 8 vanishes due to the negative $\mathring{\boldsymbol{A}}_t$, and $\boldsymbol{H}_t$ depends solely on the new token embeddings. Conversely, when $\boldsymbol{\Delta}_t$ tends to zero the second term vanishes, and $\boldsymbol{H}_t$ is entirely inherited from past memory. Therefore, $\boldsymbol{\Delta}_t$ effectively selects information from the new input token while simultaneously clearing past memory to make room for it. Note that our framework not only supports pre-training but also enables fine-tuning from a pre-trained SSM. This is achieved without introducing any additional modules—only by converting the deterministic gates into stochastic ones in a plug-and-play manner.

### 3.3 INHERENT INTERPRETABILITY

As previously discussed, each element in the state transition matrix $\boldsymbol{A}_t$ governs whether a specific component of the historical context is retained or discarded in memory (Gu et al., 2020). Preserving a larger subset of components enables the current input token to dynamically interact with and update the states of these retained components, thereby encoding critical information into the model's memory. This mechanism provides inherent interpretability to the Baleen architecture.

$$\text{Importance}(t) = \frac{1}{ND} \sum_{d=1}^{D} \sum_{n=1}^{N} \boldsymbol{A}_t^{(n,d)}, \quad t \in [T] \tag{10}$$

Table 1: Accuracy comparison on benchmark datasets. Best results are in bold, second best underlined.

| Method | ARC-C | ARC-E | BoolQ | GPQA | Hella. | MMLU | OBQA | PIQA | SIQA | TruthfulQA | Wino. | Avg. |
|---|---|---|---|---|---|---|---|---|---|---|---|---|
| **Pretrained 370M models** | | | | | | | | | | | | |
| Baleen-B | 18.94 | 36.20 | 59.39 | 24.33 | 28.02 | 22.97 | 14.80 | 59.14 | 32.91 | 23.38 | **51.22** | 33.75 |
| Baleen-E | 17.58 | **39.35** | **60.52** | 24.11 | **28.18** | 23.01 | 14.20 | **60.45** | 33.11 | 23.62 | 48.38 | **33.86** |
| Gated DeltaNet | **23.29** | 32.45 | 51.25 | 24.11 | 26.99 | 23.29 | 15.00 | 56.96 | **35.11** | 25.21 | 51.85 | 33.23 |
| Mamba2 | 17.83 | 37.54 | 49.66 | 24.33 | 27.84 | 22.92 | 14.40 | 59.96 | 33.01 | 23.87 | 50.67 | 32.91 |
| Mamba | 22.01 | 29.25 | 45.69 | **25.89** | 25.91 | **24.98** | 13.20 | 54.62 | 32.70 | 25.21 | 49.49 | 31.72 |
| RetNet | 21.67 | 30.85 | 57.13 | 24.33 | 26.50 | 23.60 | 14.20 | 55.93 | 33.93 | 24.11 | 48.78 | 32.82 |
| RWKV6 | 21.76 | 31.06 | 46.21 | 25.67 | 26.92 | 24.10 | **15.80** | 55.82 | 33.01 | **26.56** | 49.96 | 32.44 |
| GLA | 21.93 | 30.26 | 44.56 | 24.78 | 26.52 | 23.76 | **15.60** | 56.42 | 33.62 | 25.09 | 49.41 | 32.00 |
| **Finetuned 7B models** | | | | | | | | | | | | |
| Baleen-B 7B | 31.23 | 61.99 | **73.06** | 27.23 | 39.37 | **34.76** | 21.80 | 67.57 | 32.91 | **27.66** | 58.01 | **43.24** |
| Baleen-E 7B | 31.40 | 60.94 | 71.74 | 20.54 | **40.26** | 33.70 | **22.40** | **68.28** | 32.96 | 27.54 | 56.99 | 42.43 |
| Mamba2 7B | **33.19** | **62.08** | 62.63 | 27.01 | 38.06 | 32.84 | 19.80 | 67.63 | 32.91 | 27.17 | 57.14 | 41.86 |

To operationalize interpretation during inference, we first compute the expectation of the stochastic transition matrices $A_t$ and use it as state transition matrix in Eq. 1. We then derive token-level importance scores by averaging the magnitudes of the transition matrices across different components (i.e., averaging over embedding dimension $D$ and state dimension $N$ as Eq. 10 shows). These scores enable the identification of salient tokens through a top-$k$ ranking strategy, offering transparent insights into the model's decision-making process during prediction.

## 4 EXPERIMENTS

In this section, we will validate our effectiveness in common language benchmarks, providing faithful inherent interpretation, and ensuring robustness under perturbations.

**Settings.** We pre-trained 370M Baleen-B and Baleen-E models based on Mamba2 (Dao & Gu, 2024) on 20B tokens from SlimPajama dataset (Soboleva et al., 2023), and fine-tuned 7B Baleen-B and Baleen-E models from a pre-trained Mamba2 model (Codestral (AI@Mistral, 2024)) on 0.5B instruction-following formatted tokens from Crystal dataset (Liu et al., 2023). We mainly compare our method with the vallina Mamba2 model, and add other state space or linear attention models as baselines including RetNet (Sun et al., 2023), RWKV6 (Peng et al., 2023), GLA (Yang et al., 2023) Mamba (Gu & Dao, 2023), and recently proposed Gated DeltaNet (Yang et al., 2024a).

### 4.1 LANGUAGE BENCHMARKS

**Pre-trained 370M Models.** Baleen-B (Baleen-B) and Baleen-E (Baleen-E) achieve the best overall performance, with average accuracies of **33.75%** and **33.86%**, both surpassing all baselines out-performing all baselines including the strong Gated-DeltaNet (33.23%), which is the architecture behind the popular industry model Qwen3-Next (Qwen-Team, 2025). On commonsense reasoning tasks, Baleen-E leads ARC-Easy (**39.35%**, +1.81 over Mamba2) and PIQA (**60.45%**, +0.49), while Baleen-B ranks first on Winogrande (**51.22%**, +0.55). For language understanding, their MMLU (22.97% and 23.01%) scores slightly exceed Mamba2 (22.92%). Overall, both Baleen variants show consistent gains over Mamba2, confirming the benefit of our design on top of the Mamba2 architecture.

**Fine-tuned 7B Models.** After instruction tuning, Baleen-B attains the highest average accuracy of **43.24%**, followed by Baleen-E (**42.43%**), both ahead of Mamba2 (41.86%). In commonsense reasoning, Baleen-B excels on BoolQ (**73.06%**, +10.43), while Baleen-E leads PIQA (**68.28%**, +1.65). Both outperform Mamba2 on Winogrande (58.01% and 56.99% vs. 57.14%). For language understanding, Baleen-B reaches 34.76% on MMLU (+1.92), while Baleen-E achieves the highest HellaSwag (40.26%, +2.2) and competitive MMLU (33.70%) scores. On TruthfulQA, Baleen-B achieves 27.66% (+0.49), slightly above Mamba2, while Baleen-E performs slightly below Mamba2. Overall, Baleen-B and Baleen-E consistently outperform the Mamba2 baseline, delivering stronger commonsense reasoning and steady gains in language understanding and truthfulness.

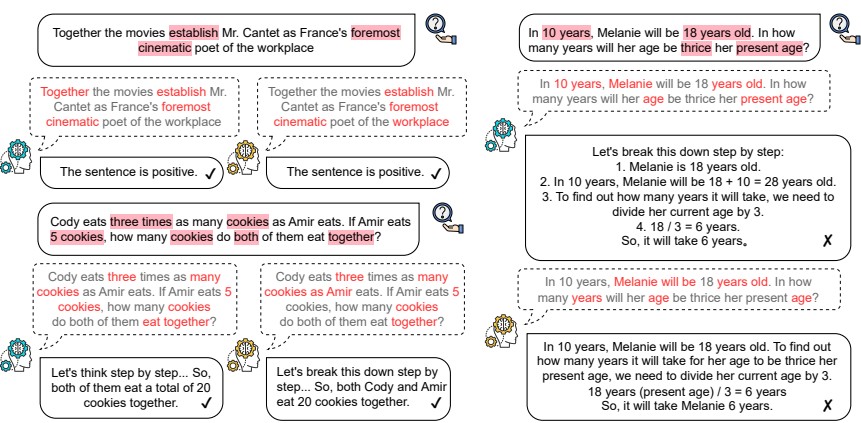

(a) Right answer interpretation. All important tokens are captured.    (b) Wrong answer interpretation. 18 and thrice are not captured.

Figure 2: Interpretation Examples on SST and GSM8K. Human annotated important tokens in questions are highlighted in red background, and model interpretation after top-k ranking ($k = 4/8$ for SST/GSM8K) are highlighted in red font. The blue avatar refers to Baleen-E 7B and yellow avatar refers to Baleen-B 7B.

## 4.2 INHERENT INTERPRETATION

**Qualitative Example** We test fine-tuned 7B model in SST and GSM8K to yield inherent interpretation when generating answer. We visualize three samples with human annotated ground truth interpretation from previous literature (Wu et al., 2023) in Fig. 2.

As Fig. 2 (a) shows, both Baleen-E (blue) and Baleen-B (yellow) are able to provide accurate interpretation aligned with human annotations well when generating correct answers. For sentiment analysis, three out of the four most important tokens indicating positive emotion are identified. For the math problem, key elements including objects, numbers, and the word "together" are selected.

In contrast, Fig. 2 (b) illustrates how the model makes a wrong answer in a math problem: It fails to properly attend to the tokens "18" and "thrice" in the input. Consequently, the model incorrectly treats Melanie's current age as "18" instead of "8", and divides the age by 3 rather than multiplying it by 3. These two mistakes are directly attributable to the model's inattention to "18" and "thrice", which clearly explains why the output is wrong. Such analysis is useful for monitoring the model's long-form text generation, enabling early identification of errors and preventing wasted computation. Moreover, it reveals potential weaknesses in the model and provides guidance for further improvement.

**Quantitative Evaluation.** We conduct quantitative evaluation regarding model interpretations and report results in Tab. 2. Fidelity AUC (Zhu et al., 2024; Jafari et al., 2024) is often adopted to reduce the evaluation bias of single sparsity value (i.e., Fidelity@k) which cannot provide consistent evaluations. We use four well-established text classification tasks SNLI (Bowman et al., 2015), IMDb (Maas et al., 2011), SST2 (Socher et al., 2013) and Rottent Tomatoes (Pang & Lee, 2005), and fine-tuned pretrained 370M model on them. We compare our methods (self-interpretation) with commonly used gradient-based post-hoc methods

Table 2: Explanation performance on two base models trained with Baleen-E and Baleen-B respectively. "RT." refers to Rotten Tomatoes.

| Method | SNLI | IMDb | SST2 | RT. |
|---|---|---|---|---|
| **Baleen-B** | | | | |
| IG | 44.00 | 40.21 | 39.94 | 40.50 |
| GradX | 48.85 | 42.98 | 47.96 | 40.94 |
| GradCAM | 54.98 | **49.59** | 42.57 | 42.00 |
| Self (Ours) | **55.81** | 45.24 | **57.65** | **51.49** |
| **Baleen-E** | | | | |
| IG | 24.23 | 2.45 | 38.47 | 2.27 |
| GradX | 48.88 | 43.69 | 47.79 | 41.63 |
| GradCAM | **54.73** | **52.69** | 39.19 | 48.65 |
| Self (Ours) | 54.31 | 51.66 | **62.50** | **62.17** |

including Gradient × Input (Shrikumar et al., 2017), Integrated Gradients (IG) (Sundararajan et al., 2017) and GradCAM (Selvaraju et al., 2017). For the fair comparison, we ensure that the interpreted model remains the same; thus, Baleen-B 370M and Baleen-E 370M serve as the base models. Notably, our method is self-interpretable and can generate interpretations without additional compute budget.

As demonstrated in Tab. 2, both Baleen-B and Baleen-E consistently generate highly faithful interpretations compared to existing methods, with their average performance significantly surpassing

Table 3: Adversarial attack results of CIFAR-10 image classification on sequences of pixels.

| Models | [0:0] | [0:32] | [0:96] | [928:1024] | [992:1024] | Avg. |
|---|---|---|---|---|---|---|
| | | | **Perturbation region** (seq. length = 1024) | | | |
| RWKV | 0.474 | 0.150 (↓ 68.35%) | 0.466 (↓ **1.58%**) | 0.138 (↓ 70.88%) | 0.460 (↓ **2.91%**) | 0.338±0.177 |
| GLA | 0.631 | 0.623 (↓ 1.42%) | 0.597 (↓ 5.41%) | 0.332 (↓ 47.36%) | **0.496** (↓ 21.42%) | 0.536±0.126 |
| Mamba | **0.672** | **0.655** (↓ 2.54%) | **0.631** (↓ 6.19%) | 0.102 (↓ 84.77%) | 0.152 (↓ 77.35%) | 0.442±0.289 |
| Mamba2 | **0.673** | 0.495 (↓ 26.51%) | 0.470 (↓ 30.17%) | 0.106 (↓ 84.17%) | 0.183 (↓ 72.74%) | 0.385±0.235 |
| Baleen-B | 0.659 | 0.634 (↓ 3.82%) | 0.603 (↓ 8.58%) | 0.314 (↓ 52.37%) | 0.453 (↓ 31.23%) | 0.532±0.146 |
| Baleen-E | 0.643 | 0.639 (↓ **0.57%**) | 0.628 (↓ 2.33%) | **0.365** (↓ 43.16%) | 0.433 (↓ 32.63%) | **0.542±0.132** |

baseline approaches. This inherent self-interpretability enables real-time monitoring of model behavior, allowing practitioners to promptly detect errors during inference rather than relying on post hoc analysis.

## 4.3 ADVERSARIAL ROBUSTNESS

**Position Bias Attack.** SSMs exhibit position bias, which can be utilized to compromise their performance (Wang et al., 2024). To assess whether our method mitigates this bias, we evaluate model robustness against corrupted inputs through *image classification on sequences of pixels* (Tay et al., 2020). For this task, each $W \times H$ image is flattened into a sequence of $WH$ RGB pixel tokens, which are then embedded into the hidden state dimension and processed by sequence modeling blocks. Following ViT (Dosovitskiy et al., 2021), we append a learnable class token at the end of the sequence. This token is subsequently mapped to logits via a classifier head for image classification. This design introduces a vulnerability: since predictions rely on the class token, position bias implies that tokens closer to it exert a greater influence on the output. Consequently, SSMs with weaker robustness are particularly vulnerable to perturbations in trailing tokens near the class token. To highlight this bias, we generate corrupted inputs by perturbing both leading and trailing tokens.

In our image classification experiments on CIFAR-10 (Tab. 3), we find both our variants demonstrate balanced sensitivity to perturbations at leading and trailing positions. Under leading-token corruption (`[0:32]`), Baleen-B and Baleen-E retain 0.634 accuracy (3.8% drop) and 0.639 (0.6% drop), respectively, maintaining moderate performance degradation comparable to the original Mamba while outperforming the base model Mamba2. However, trailing-token corruption (`[992:1024]`) leads to catastrophic accuracy losses for both Mamba and Mamba2 (>70%). In contrast, SSIBs demonstrate significantly improved robustness, limiting accuracy drops to approximately 30% under the same conditions. Notably, Baleen-B achieves the highest average accuracy across all compared methods.

**Prompt Attack.** To further evaluate the robustness improvement by our method, we conduct prompt attack experiments on our fine-tuned 7B language models using WNLI and SST2 dataset (Wang et al., 2018). Following the setup in PromptBench (Zhu et al., 2023a), we apply four different types of perturbation: DeepWordBug, TextBugger, CheckList, and StressTest (Zhu et al., 2023b) varing from character-level attack to sentence-level attack. Robustness is assessed by measuring the performance drop ratio (PDR) in prediction accuracy between normal and perturbed prompts.

Tab. 4 compares the performance degradation of our fine-tuned 7B Baleen-E and Baleen-B models against 7B Mamba2. As shown, both Baleen-E and Baleen-B consistently achieve lower drop ratios across most adversarial settings, indicating stronger robustness to perturbations. In particular, Baleen-B demonstrates superior resilience on SST2 under Checklist and StressTest, with performance drop ratios that are 44.66% and 18.55% lower than those of Mamba2, respectively. Meanwhile, Baleen-E shows competitive robustness under DeepWordBug and TextBugger, achieving drop ratios that are 32.14% and 18.16% lower than Mamba2. On WNLI, Baleen-E substantially outperforms the other models on Checklist and TextBugger, achieving drop ratios as low as 17.65 and 0.00 respectively. These results highlight that our fine-tuned Baleen models not only mitigate performance degradation more effectively than Mamba2 but also complement each other in handling different types of adversarial attacks.

## 5 RELATED WORK

**Linear State Space Modeling as Online Learning.** Previous works (Sun et al., 2024; Liu et al., 2024; Behrouz et al., 2024) have proposed to unify the computational structure of linear SSMs from

Table 4: Performance drop ratio (lower is better) results after adversarial attack on PromptBench.

| Model | SST2 | | | | WNLI | | | |
|---|---|---|---|---|---|---|---|---|
| | Checklist | DeepWordBug | StressTest | TextBugger | Checklist | DeepWordBug | StressTest | TextBugger |
| Baleen-B | **32.26** | 43.16 | **43.48** | 71.58 | 61.76 | **6.67** | 94.74 | 9.38 |
| Baleen-E | 39.08 | **38.37** | 71.26 | **70.45** | **17.65** | 14.71 | **87.50** | **0.00** |
| Mamba2 | 76.92 | 70.51 | 62.03 | 88.61 | 80.00 | 13.33 | 100.00 | 10.00 |

an online learning perspective. The update rule of SSMs can be generally understood as doing a single step of online stochastic gradient descent, with respect to the state spaces $\mathbf{H}_t$, for optimizing an objective $L(\mathbf{H}_t) = ||\mathbf{H}_t - \alpha_t \mathbf{H}_{t-1}||$ at the time stamp $t$ to control the magnitude of the state update. These online learning objectives often introduce additional regularization terms for better modeling the key-value associations, resulting in different variants of state updates consisting of gating (Sun et al., 2023; Gu & Dao, 2023; Yang et al., 2023; Peng et al., 2023; De et al., 2024; Goldstein et al., 2024), delta rule (Schlag et al., 2021; Yang et al., 2024b;a) and momentum (Behrouz et al., 2024). Our approach is loosely connected to this perspective, in the sense that we regard the hidden states as an information bottleneck between the input and the target sequences, and the KL term functions as a regularizer of the hidden state $\mathbf{H}_t$ with respect to a prior distribution $\mathbb{Q}$. Since Linear Attention (Qin et al., 2022; Sun et al., 2023) can be viewed as SSMs with linear recurrent updates of two-dimensional associative memories, our approach can be applied to a broader spectrum of architectures (MiniMax et al., 2025; Ren et al., 2023; 2024; Lieber et al., 2024; Ma et al., 2022; 2024) consisting of linear recurrent models.

**Interpretation Methods.** Previous interpretation methods can form into two categories (Arrieta et al., 2020; Rudin, 2019). The first category, known as post-hoc methods, operates on already-trained models. For instance, Gradient $\times$ Input (Shrikumar et al., 2017) compute the product of the input and the gradients w.r.t. the input as the importance scores. GradCAM (Selvaraju et al., 2017) generalize the operations to intermediate embeddings and the corresponding gradients. The second category comprises self-interpretable models. These models integrate interpretable modules into model architectures, rooted in principled mechanism such as causality (Wu et al., 2022), information bottleneck (Jiang et al., 2020) or and even attention mechanism (Smilkov et al., 2017). These models are specifically designed to extract and utilize minimal significant information during model training. For instance, vanilla attention score used to be treat as self-interpretation, but are proved to be not faithful (Jain & Wallace, 2019), and then improved via modified attention mechanism (Chrysostomou & Aletras, 2021) and interpretation deduction (Liu et al., 2022). Our method belongs to self-interpretable methods with theoretical principle.

**Adversarial Robustness.** As large language models (LLMs) increasingly dominate natural language processing and permeate daily life through their conversational capabilities, concerns about their reliability and alignment with human values have grown. Robustness is a key factor, requiring models to remain reliable even under adversarial or challenging conditions. Existing approaches mostly rely on post-hoc defenses such as detectors or input/output filters (Dong et al., 2024), but more subtle attacks (e.g., noisy data injection during training) highlight the need for inherently robust architectures. Prior studies explore robustness enhancements through mutual information regularization (Wang et al.), information bottleneck methods (Zhang et al., 2022), or selective fine-tuning (Kim et al., 2023), though these works mainly target smaller models like BERT or RoBERTa. To the best of our knowledge, we are the first to demonstrate robustness at the scale of modern LLMs (e.g., 7B parameters) using a scalable SSM-based architecture.

## 6 CONCLUSION

In this work, we introduce *Baleen*, a family of SSM architectures that explicitly maximizes context compression with reliable interpretation based on *State Space Information Bottleneck* framework. It is realized by stochastic modeling of the selection mechanism, enforces memory states to be *minimal and sufficient* representation of the past context. Extensive experiments show Baleen addresses the limitations of existing SSMs in selectivity, robustness, and interpretability.

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

## A  DEFERRED THEORY

In this section, we provide detailed derivations for SSIB framework with two types of stochasticity.

### A.1  BACKGROUND

In this section, we provide readers with basic concepts in information theory. Below our introduction focuses on two random variables $A, B$. Extending the below definitions to random vectors/matrices is straightforward. Interested readers are referred to Cover (1999).

**Entropy.** The entropy of a random variable $A$ quantifies how much information it carries in its distribution, often characterized by how many "bits" are needed to represent the random variable. It is defined as below:

$$\mathbb{H}(A) = -\mathbb{E}_A\left[\log\mathbb{P}(A)\right]. \tag{11}$$

Higher entropy indicates greater uncertainty in the possible values of $A$.

**Mutual Information.** The mutual information between two random variables $A$ and $B$ measures the amount of information $A$ provides about $B$:

$$\mathbb{I}(A;B) = \mathbb{I}(B;A) = \mathbb{E}_{A,B}\left[\log\frac{\mathbb{P}(A,B)}{\mathbb{P}(A)\mathbb{P}(B)}\right]. \tag{12}$$

It captures the reduction in uncertainty of one variable given knowledge of the other.

**KL Divergence.** The Kullback-Leibler (KL) divergence measures how different a probability distribution $\mathbb{P}(A)$ is from a reference distribution $\mathbb{Q}(A)$:

$$D_{\mathrm{KL}}(\mathbb{P}(A)\|\mathbb{Q}(A)) = \mathbb{E}_{A\sim\mathbb{P}}\left[\log\frac{\mathbb{P}(A)}{\mathbb{Q}(A)}\right]. \tag{13}$$

KL divergence is always non-negative and equals zero if and only if $\mathbb{P}(A) = \mathbb{Q}(A)$ almost everywhere.

As seen in Sec. 3, these information-theoretic metrics play a fundamental role in the information bottleneck framework. We assume all above quantities are well-defined throughout this paper. They are also closely related to each other. Below are some useful equalities among entropy, mutual information, and KL divergence:

$$\mathbb{H}(A|B) = \mathbb{H}(A,B) - \mathbb{H}(B) \tag{14}$$

$$\mathbb{I}(A;B) = \mathbb{H}(A) - \mathbb{H}(A|B) = \mathbb{H}(B) - \mathbb{H}(B|A) = \mathbb{H}(A) + \mathbb{H}(B) - \mathbb{H}(A,B), \tag{15}$$

$$\mathbb{I}(A;B) = D_{\mathrm{KL}}(\mathbb{P}(A,B)\|\mathbb{P}(A)\mathbb{P}(B)). \tag{16}$$

## A.2 SSIB TRAINING OBJECTIVE

In this section, we derive the main results in Sec. 3.1. We begin by specifying the PDFs for inputs $\boldsymbol{X}$, memory states $\boldsymbol{H}$, and ground-truth sequence $\boldsymbol{Y}$. First of all, $\mathbb{P}(\boldsymbol{X})$ or $\mathbb{P}_{\boldsymbol{X}}$ is an unknown distribution where input sequences are sampled from. The conditional distribution $\mathbb{P}(\boldsymbol{Y}|\boldsymbol{X})$ or $\mathbb{P}_{\boldsymbol{Y}|\boldsymbol{X}}$ is another unknown data distribution describing how ground-truth sequences are generated from $\boldsymbol{X}$. We use contextual encoder $h_\Phi$ to extract information from $\boldsymbol{X}$, characterizing by another conditional distribution $\mathbb{P}^\Phi(\boldsymbol{H}|\boldsymbol{X})$ or $\mathbb{P}^\Phi_{\boldsymbol{H}|\boldsymbol{X}}$. The predictor $g_\Theta$ yields a conditional probability $\mathbb{P}^\Theta(\boldsymbol{Y}|\boldsymbol{H})$ or $\mathbb{P}^\Theta_{\boldsymbol{Y}|\boldsymbol{H}}$, describing the likelihood estimation of groundtruth $\boldsymbol{Y}$ under memory $\boldsymbol{H}$ and parameter $\Theta$.

**A Tractable Objective.** We begin with restating the objective of the SSIB framework in Sec. 5:

$$\min_{\Theta,\Phi} -\mathbb{I}(\boldsymbol{Y};\boldsymbol{H}) + \beta\mathbb{I}(\boldsymbol{H};\boldsymbol{X}), \quad \text{s.t. } \boldsymbol{H} \sim \mathbb{P}^\Phi(\boldsymbol{H}|\boldsymbol{X}). \tag{17}$$

The mutual information terms in Eq. 17 is not computationally tractable. Hence, we seek to obtain an upper bound for it following the approaches in Alemi et al. (2016); Poole et al. (2019).

First, we aim to lower bound $\mathbb{I}(\boldsymbol{Y};\boldsymbol{H})$. Note that $\mathbb{P}(\boldsymbol{Y}|\boldsymbol{H})$ is not tractable, so we introduce a variational approximation $\mathbb{P}^\Theta(\boldsymbol{Y}|\boldsymbol{H})$ for it, leading to:

$$\mathbb{I}(\boldsymbol{H};\boldsymbol{Y}) = \mathbb{E}_{\boldsymbol{H},\boldsymbol{Y}}\left[\log\frac{\mathbb{P}^\Theta(\boldsymbol{Y}|\boldsymbol{H})}{\mathbb{P}(\boldsymbol{Y})}\right] + \mathbb{E}_{\boldsymbol{H}}\left[D_{\mathrm{KL}}(\mathbb{P}(\boldsymbol{Y}|\boldsymbol{H})\|\mathbb{P}^\Theta(\boldsymbol{Y}|\boldsymbol{H}))\right] \tag{18}$$

$$\geq \mathbb{E}_{\boldsymbol{H},\boldsymbol{Y}}\left[\log\frac{\mathbb{P}^\Theta(\boldsymbol{Y}|\boldsymbol{H})}{\mathbb{P}(\boldsymbol{Y})}\right] \tag{19}$$

$$= \mathbb{E}_{\boldsymbol{H},\boldsymbol{Y}}\left[\log\mathbb{P}^\Theta(\boldsymbol{Y}|\boldsymbol{H})\right] + \mathbb{H}(\boldsymbol{Y}). \tag{20}$$

The first inequality is according to the non-negativity of KL divergence. Thus, $\mathbb{I}(\boldsymbol{H};\boldsymbol{Y})$ is lower bounded by the expected log-likelihood plus the entropy of $\boldsymbol{Y}$. The log-likelihood term of Eq. 20

corresponds to the prediction step in SSMs ($g_\Theta$ and Eq. 1). The entropy term is independent of $\Theta$ and will be ignored in our final objective. Moreover, note that the predicted variable is independent of other variables given $\boldsymbol{H}_t$: $\mathbb{P}^\Theta(\boldsymbol{Y}|\boldsymbol{H}) = \prod_t \mathbb{P}^\Theta(\boldsymbol{Y}_t|\boldsymbol{H}_t)$. Therefore, we are able to further expand Eq. 20 as:

$$\mathbb{I}(\boldsymbol{H}; \boldsymbol{Y}) \geq \mathbb{E}_{\boldsymbol{H},\boldsymbol{Y}} \left[ \sum_{t=1}^T \log \mathbb{P}^\Theta(\boldsymbol{y}_t|\boldsymbol{H}_t) \right] + \mathbb{H}(\boldsymbol{Y}). \tag{21}$$

Second, we upper bound for $\mathbb{I}(\boldsymbol{H}; \boldsymbol{X})$. By definition, the mutual information can be written as:

$$\mathbb{I}(\boldsymbol{H}; \boldsymbol{X}) = \mathbb{E}_{\boldsymbol{H},\boldsymbol{X}} \left[ \log \frac{\mathbb{P}^\Phi(\boldsymbol{H}|\boldsymbol{X})}{\mathbb{P}^\Phi(\boldsymbol{H})} \right], \tag{22}$$

where $\mathbb{P}(\boldsymbol{H}) = \mathbb{E}_{\boldsymbol{X}}[\mathbb{P}^\Phi(\boldsymbol{H}|\boldsymbol{X})]$ is again intractable. We borrow the technique of Kingma (2013) and introduce a variational approximation $\mathbb{Q}(\boldsymbol{H})$, resulting in:

$$\mathbb{I}(\boldsymbol{H}; \boldsymbol{X}) = \mathbb{E}_{\boldsymbol{H},\boldsymbol{X}} \left[ \log \frac{\mathbb{P}^\Phi(\boldsymbol{H}|\boldsymbol{X})}{\mathbb{Q}(\boldsymbol{H})} \right] - D_{\mathrm{KL}}(\mathbb{P}(\boldsymbol{H})\|\mathbb{Q}(\boldsymbol{H})) \tag{23}$$

$$\leq \mathbb{E}_{\boldsymbol{X}} \left[ D_{\mathrm{KL}}(\mathbb{P}^\Phi(\boldsymbol{H}|\boldsymbol{X})\|\mathbb{Q}(\boldsymbol{H})) \right], \tag{24}$$

which implies $\mathbb{I}(\boldsymbol{H}; \boldsymbol{X})$ is upper bounded by the expected KL divergence between the variational approximation and the marginalized distribution. The inequality becomes tighter if $\mathbb{Q}(\boldsymbol{H})$ approximates $\mathbb{P}^\Phi(\boldsymbol{H})$ closely. $\mathbb{P}^\Phi(\boldsymbol{H}|\boldsymbol{X})$ in the RHS of Eq. 24 functions as the encoding step in SSMs ($h_\Phi$ and Eq. 1).

Combining both Eq. 24 and 20, we can recover Eq. 5:

$$\min_{\Theta,\Phi} - \mathbb{E}_{\boldsymbol{H},\boldsymbol{Y}} \left[ \sum_{t=1}^T \log \mathbb{P}^\Theta(\boldsymbol{y}_t|\boldsymbol{H}_t) \right] + \beta\, \mathbb{E}_{\boldsymbol{X}} \left[ D_{\mathrm{KL}}(\mathbb{P}^\Phi(\boldsymbol{H}|\boldsymbol{X})\|\mathbb{Q}(\boldsymbol{H})) \right], \tag{25}$$

**Derivation of KL Divergence Constraints.** In this section, we simplify the KL divergence under the setting of Baleen-B. The main idea is to utilize the Markov property and factorize $\boldsymbol{H}$ along the sequence.

Last, let us fix $\boldsymbol{X}$ and simplify the KL divergence term:

$$D_{\mathrm{KL}}(\mathbb{P}^\Phi(\boldsymbol{H}|\boldsymbol{X})\|\mathbb{Q}(\boldsymbol{H})) = \mathbb{E}_{\boldsymbol{H}} \left[ \log \frac{\mathbb{P}^\Phi(\boldsymbol{H}|\boldsymbol{X})}{\mathbb{Q}(\boldsymbol{H})} \right] \tag{26}$$

$$= \mathbb{E}_{\boldsymbol{H}} \left[ \log \frac{\mathbb{P}^\Phi(\boldsymbol{H}_0) \prod_{t=1}^T \mathbb{P}^\Phi(\boldsymbol{H}_t|\boldsymbol{H}_{t-1}, \boldsymbol{X}_{\leq t})}{\mathbb{Q}(\boldsymbol{H}_0) \prod_{t=1}^T \mathbb{Q}(\boldsymbol{H}_t|\boldsymbol{H}_{t-1})} \right] \tag{27}$$

$$= \mathbb{E}_{\boldsymbol{H}} \left[ \log \frac{\mathbb{P}^\Phi(\boldsymbol{H}_0)}{\mathbb{Q}(\boldsymbol{H}_0)} + \sum_{t=1}^T \log \frac{\mathbb{P}^\Phi(\boldsymbol{H}_t|\boldsymbol{H}_{t-1}, \boldsymbol{X}_{\leq t})}{\mathbb{Q}(\boldsymbol{H}_t|\boldsymbol{H}_{t-1})} \right] \tag{28}$$

$$= \mathbb{E}_{\boldsymbol{H}_0} \left[ \log \frac{\mathbb{P}^\Phi(\boldsymbol{H}_0)}{\mathbb{Q}(\boldsymbol{H}_0)} \right] + \sum_{t=1}^T \mathbb{E}_{\boldsymbol{H}} \left[ \log \frac{\mathbb{P}^\Phi(\boldsymbol{H}_t|\boldsymbol{H}_{t-1}, \boldsymbol{X}_{\leq t})}{\mathbb{Q}(\boldsymbol{H}_t|\boldsymbol{H}_{t-1})} \right] \tag{29}$$

$$= \sum_{t=1}^T \mathbb{E}_{\boldsymbol{H}} \left[ \log \frac{\mathbb{P}^\Phi(\boldsymbol{H}_t|\boldsymbol{H}_{t-1}, \boldsymbol{X}_{\leq t})}{\mathbb{Q}(\boldsymbol{H}_t|\boldsymbol{H}_{t-1})} \right] + const., \tag{30}$$

where the last equality holds because $\mathbb{P}^\Phi(\boldsymbol{H}_0)$ is independent of model parameters.

## A.3 SIMPLIFIED KL LOSS IN BALEEN

**Baleen-E** Given prior $\mathbb{Q}(A_t)$ and $\mathbb{P}(A_t)$, and equation $H_t = H_{t-1}\exp(\mathring{A}_t\mathrm{diag}(\Delta_t)) + B_t x_t \Delta_t$. We can introduce the conditional distribution $\Delta_t|H_t$ and use the KL divergence chain rule:

$$D_{\mathrm{KL}}(\mathbb{P}(\Delta_t)\|\mathbb{Q}(\Delta_t)) = \mathbb{E}_{\boldsymbol{\Delta_t}}\left[\log \frac{\mathbb{P}(\Delta_t\|H_t)\mathbb{P}(H_t|H_{t-1})}{\mathbb{Q}(\Delta_t\|H_t)\mathbb{Q}(H_t)}\right] \tag{31}$$

$$= \mathbb{E}_{\boldsymbol{\Delta_t}|\boldsymbol{H_t}}\,\mathbb{E}_{\boldsymbol{H_t}}\left[\log \frac{\mathbb{P}(H_t|H_{t-1})}{\mathbb{Q}(H_t)}\right] + \mathbb{E}_{\boldsymbol{H_t}}\,\mathbb{E}_{\boldsymbol{\Delta_t}|\boldsymbol{H_t}}\left[\log \frac{\mathbb{P}(\Delta_t\|H_t)}{\mathbb{Q}(\Delta_t\|H_t)}\right] \tag{32}$$

$$= D_{\mathrm{KL}}(\mathbb{P}(H_t|H_{t-1})\|\mathbb{Q}(H_t) + D_{\mathrm{KL}}(\mathbb{P}(\Delta_t|H_t)\|\mathbb{Q}(\Delta_t|H_t) \tag{33}$$

$$\geq D_{\mathrm{KL}}(\mathbb{P}(H_t|H_{t-1})\|\mathbb{Q}(H_t) \tag{34}$$

**Baleen-B**  Given prior $\mathbb{Q}(A_t)$ and $\mathbb{P}(A_t)$, and equation $H_t = H_{t-1}A_t + B_t x_t$, we obtain the following relation, analogous to the derivation in Baleen-B:

$$D_{\mathrm{KL}}(\mathbb{P}(A)\|\mathbb{Q}(A_t)) = D_{\mathrm{KL}}(\mathbb{P}(H_t|H_{t-1})\|\mathbb{Q}(H_t)) + D_{\mathrm{KL}}(\mathbb{P}(A_t|H_t)\|\mathbb{Q}(A_t|H_t)) \tag{35}$$

$$= D_{\mathrm{KL}}(\mathbb{P}(H_t|H_{t-1})\|\mathbb{Q}(H_t)) \tag{36}$$

In this case, the inequality becomes equality because $D_{\mathrm{KL}}(\mathbb{P}(A_t|H_t)\|\mathbb{Q}(A_t|H_t) = 0$ which follows directly from the one-to-one mapping between $A_t$ and $H_t$ given $H_{t-1}$.

# B  EXPERIMENT DETAILS

## B.1  ADVERSARIAL ROBUSTNESS.

**Position Bias Attack.**  In this experiment, we train all the compared models using the framework from (Arora et al., 2023). Each model consists of three layers, with a hidden state dimension of 32 per layer. Training is conducted for 100 epochs on the CIFAR-10 dataset. For our Baleen-B model, we set $\beta = 0.01$, while for the Baleen-E model, $\beta = 0.5$.

**Prompt Attack.**  The following text prompt is used to query the language models in this experiment, where the placeholder '{content}' will be replaced with two sentences. The second sentence provides an explanation of the pronoun used in the first sentence. The language model is then tasked with predicting whether the pronoun resolution is "correct" or "incorrect."

> In the following sentence, does the hypothesis correctly resolve the ambiguous pronoun? Answer 'correct' or 'incorrect'. Please classify:
> Question: {content}
> Answer:

Below is a negative sample of the two sentences:

> Sentence 1: The drain is clogged with hair. It has to be cleaned.
> Sentence 2: The hair has to be cleaned.

