# OpenReview forum: "Baleen: Self‑Interpretable, Robust SSMs with Stochastic Selective Memory"
_ICLR.cc/2026/Conference — Submitted to ICLR 2026_

### Official Review · Reviewer_yxmK · 2025-10-21

**Soundness:** 4
**Presentation:** 4
**Contribution:** 4
**Rating:** 6
**Confidence:** 1

**Summary:**

This paper introduces an improvement to the state space model, introducing Beleen families. This is attributed to a careful analysis of Mamba from the aspect of the information bottleneck. Baleen studies the selection mechanism from Mamba and has two new variants of the selection strategy. Experiments show the performance enhancement brought about by the new selection methods. Other aspects, including interpretation and performance robustness, are also studied.

**Strengths:**

- Clear motivation: This paper clearly revisits the information-theoretic pitfalls of Mamba and proposes the improved Baleen architecture.
- Plug-and-play designs: The proposed designs can be incorporated into existing Mamba models.
- Comprehensive experiments: Various aspects of the Balle model are investigated, including language benchmarks, inherent interpretation, and robustness.

**Weaknesses:**

- Inexact terms: Section 4.3 discusses the model’s performance robustness to perturbations. This does not constitute an adversarial scenario, as there is no clear attacker or malicious intent.
- Missing baselines: For the language benchmark, what about the performance of transformer-based models of equal size?

**Questions:**

Will the authors release their codes to reproduce the experimental results?

---

> ### Author Response · Authors · 2025-11-28
>
> Thank the reviewer for the insightful feedback. Responses are as below.
>
> > Weakness 1. Inexact terms: Section 4.3 discusses the model’s performance robustness to perturbations. This does not constitute an adversarial scenario, as there is no clear attacker or malicious intent.
>
> In Section 4.3, the goal of the position-bias attack is to induce misclassification, and the goal of the prompt attack is to cause the model to produce incorrect answers. Under same settings, the term “adversarial attack” is used in prior work [1,2].
>
>
> > Weakness 2. Missing baselines: For the language benchmark, what about the performance of transformer-based models of equal size?
>
> Modern LLaMA-style Transformers are always stronger than purely linear models, which is expected given their higher compute (roughly >1.5× more for the same sequence length [3]). But Gated DeltaNet (GDN) already demonstrates that hybrid GDN–attention models can outperform strong Transformer baselines, and the rankings of linear models are strongly correlated their hybrid counterpart; since Baleen outperforms both Mamba2 and GDN in the pure SSM regime, we expect hybrid “Baleen + attention” variants to be stronger than these Transformer and hybrid baselines.
>
> > Question. Will the authors release their codes to reproduce the experimental results?
>
> Yes, we will open-source code and data.
>
> ---
>
> **References**:
>
> [1] Peihao Wang et al., Understanding and Mitigating Bottlenecks of State Space Models through the Lens of Recency and Over-smoothing, ICLR 2025.
>
> [2] Kaijie Zhu et al., A Unified Library for Evaluation of Large Language Models, arXiv:2312.07910, 2023.
>
> [3] Wang et al., Linformer: Self-Attention with Linear Complexity, 2020.

---

### Official Review · Reviewer_vu9z · 2025-10-31

**Soundness:** 3
**Presentation:** 3
**Contribution:** 2
**Rating:** 4
**Confidence:** 3

**Summary:**

This paper implements a variant of state-space models inspired by the information bottleneck principle. Thy argue that this information bottleneck principle can result in an inherently robust and interpretable architecture, whereas standard MAMBA does not have any incentives to select only useful parts of the context and can "overfit" in various ways with longer contexts. They propose a variant of the infromation bottleneck for the sequence modelling problem and provide a variational upper-bound which can be realized via stochastic transition dynamics in Mamba. They provide two formulations:  Baleen-B samples transitions from a bernoulli distributions and Bernoulli E samples the time-interval parameter randomly. They show that Baleen improves language modelling performance relative to MAMBA, can yield inherent interpretability, and appears to induce superior robustness to adversarial attacks/perturbations.

**Strengths:**

I think that this paper attacks an important problem facing state-space models: their seeming difficulty in rejecting irrelevant historical information in Mamba models. This has, for example, been documented in [1] where they attribute that a inability to learn to forget information contributes to performance degradations when Mamba models are tested beyond their training context size. Additionally, I find that the connection to information bottleneck principle is interesting and they clearly explain the derivations along the way to their variational upper bound and training algorithm. I believe that it is interesting that they derive two possible instantiations of their approach based off Bernoulli sampling of the transition matrix and . They also perform evaluations on a range of model properties, including interpretability, robustness, and general performance -- which I find are interesting.
[1] Chen, Yingfa, et al. "Stuffed Mamba: Oversized States Lead to the Inability to Forget." arXiv preprint arXiv:2410.07145 (2024).

**Weaknesses:**

**Motivations for the Method Could Be Presented Better**: The authors introduce their problem, in my opinion, far too abstractly. They comment on there being "no incentive" for the MAMBA model to compress irrelevant contextual information out. They provide some vague intuitions that this may be connected to worse performance/adversarial vulnerability etc. However, these claims are largely intuitive and do not seem entirely convincing.  I believe that the authors could better motivate their methods by discussing more direct evidence that MAMBA models detrimentally fail to remove knowledge. For example, connecting with the observations in [1] and/or providing more detailed results tying together the inability to forget with adversarial vulnerability/overfitting could be very beneficial for this paper.

**Better Clarity in the Implementations** Although the authors do a good job of presenting the mathematical derivations behind their method, the implementation details could be more clear. In fact, I would argue that too much time is spent on the mathematical derivation, relative to implementation details. After reading the paper, for example, I was a bit confused on several implementation details. For example, it was not clear to me how is inference performed in Baleen? The authors mention that the expectation of the stochastic matrices is used for interpretation, but it remained unclear if this is for the general forward pass or just the interpretations. Similarly, how are the hyperparameters for the prior distributions chosen? This seems to be important for the reproducability of the work but is highly unclear from the paper.

**Show Results Across Architecture and Sequence Length** The authors performed some experiments at a 20B token scale and superior performance along some baselines. However, I would be interested to know more about whether these benefits only arrive in a small-scale training setting. For example, in [1] they note that the relationship between the state parameters and the sequence length control how much forgetting is learned. Thus, it would be interesting to learn more about whether the benefits of BALEEN are amplified or mitigated depending on the relationship between Architecture v.s. Sequence Length. It would be good to haver more clarity about in what regimes the information bottleneck is particularly helpful, as this would improve our understanding of the method and its benefits.


**Provide Qualitative Examples of the Difference Between Standard Mamba and BALEEN Interpretations**  Currently, the paper provides some examples. of the BALEEN interpretability an argues that they are superior to Mamba. There are some quantitative results on this as well. However, I feel that the it would help my understanding of the paper if there were qualitative examples of where a comparable setup with Mamba fails.


[1] Chen, Yingfa, et al. "Stuffed Mamba: Oversized States Lead to the Inability to Forget." arXiv preprint arXiv:2410.07145 (2024).

**Questions:**

(1) Could the authors please clarify whether their problem statement is related to that shown in the Stuffed Mamba paper?
(2) Could the authors clarify the hyper-parameter tuning details for the prior parameters?
(3) Could the authors mention how is stochasticity of the parameters utilized at inference time.
(4) Could the authors mention how they expect the benefits of BALEEN to scale at different sequence lengths/state sizes?

---

> ### Author Response · Authors · 2025-11-28
>
> Thank the reviewer for the insightful feedback. Responses are as below.
>
> > Q1: Could the authors please clarify whether their problem statement is related to that shown in the Stuffed Mamba paper?
>
> Thank you for pointing out this valuable related work. We believe the core motivation of Stuffed Mamba is indeed same to ours: both address Mamba’s limited ability to “forget” old information. Our work provides an intuitive explanation from an information-theoretic perspective and proposes an architectural solution, while Stuffed Mamba offers empirical evidence that Mamba requires longer training sequence lengths in order to effectively forget.
>
> > Q2: Could the authors clarify the hyper-parameter tuning details for the prior parameters?
>
> We find Baleen is not very sensitive to β and simply fix β = 0.1 for Baleen-B and β = 1e−5 for Baleen-E without hyper-parameter tuning. These values are chosen so that the KL term has a similar magnitude to the cross-entropy loss. We will add ablation results in the appendix to further support this insensitivity.
>
> > Q3: Could the authors mention how is stochasticity of the parameters utilized at inference time.
>
> In inference time, we compute expectation of the stochastic gates for stable results across repeated runs. This is mentioned in line  283 and we will emphasize this more in next revision for clarity.
>
> > Q4: Could the authors mention how they expect the benefits of BALEEN to scale at different sequence lengths/state sizes?
>
> Thank you for raising this interesting point. Stuffed Mamba observes that the “forgetting” sequence length grows approximately linearly with the state size: smaller state sizes require shorter sequences before information is effectively forgotten. Since our method also explicitly encourages the model to memorize and forget, it can be viewed as effectively shrinking the usable state size, thereby shortening the forgetting sequence length. In other words, Baleen can be interpreted as compressing the state size by some proportion, so we expect its benefits to also scale approximately linearly with the sequence length and state size.
>
> ---
>
> > Weakness: Motivations for the Method Could Be Presented Better & Better Clarity in the Implementations.
>
> Thank you for this insightful suggestion. Both issues are important, and we will address them carefully in the revised version.
>
> > Weakness: Show Results Across Architecture and Sequence Length.
>
> This is an interesting point, and we will add experiments across architectures and sequence lengths (as discussed in our response to Q4) to support our analysis.
>
> > Weakness: Provide Qualitative Examples of the Difference Between Standard Mamba and BALEEN Interpretations.
>
> We agree this would be useful. While Mamba was not originally claimed to be self-interpretable, prior work [1] has identified possible interpretable mechanisms inside. However, it requires additional computation during the forward pass, making them less efficient than Baleen, whose interpretations are inherent to the model.
>
> **References**:
>
> [1] Ali, A., Zimerman, I., & Wolf, L. (2025). The Hidden Attention of Mamba Models. In Proceedings of the 63rd Annual Meeting of the Association for Computational Linguistics (ACL 2025), pp. 1516-1534.

---

### Official Review · Reviewer_BgmA · 2025-11-01

**Soundness:** 2
**Presentation:** 3
**Contribution:** 3
**Rating:** 4
**Confidence:** 3

**Summary:**

This paper proposes Baleen, a new state space model that combines information bottleneck principles with stochastic variables to improve interpretability and robustness in SSMs. The authors propose two variants: Baleen-B, which models the state transition matrix $A_t$ as a Bernoulli RV, and Baleen-E, which models the discretization parameter $\Delta_t$ as an exponential RV. Both variants include analytical KL divergence terms (or bounds) as regularizers.
Experiments demonstrate improvements over Mamba-2 on language benchmarks using 370M trained from scratch on 20B tokens, and 7B finetuned on 0.5B tokens. Baleen models also show better adversarial robustness, and self-interpretation capabilities derived from the transition matrix $A_t$.

**Strengths:**

Things that I liked in this paper include:

- Casting SSMs as an information-bottlenecked sequence model (Eq. 5) is a clean way to show that MLE-trained Mamba can suffer with spurious context and is sensitive to perturbations. I think the paper connects Mamba to the special case (\beta = 0) and builds from there. This provides a principled framework for understanding the trade-off between retention and compression in linear recurrent models.

- The Exponential-gated timestep (indirectly on both (A_t) and (B_t)) is well-motivated and look less ad-hoc.

- I think the adversarial robustness setting is an interesting angle to SSMs, and Baleen seems to handle that well. For example, the position bias attack experiments (Table 3) show convincing improvements, with Baleen-E maintaining a higher accuracy than Mamba-2. The prompt attack experiments also demonstrate consistent robustness improvements.

- The method is easy to implement and cause re-use existing efficient kernels. This makes Baleen a easy drop-in improvement for Mamba and also other linear recurrent models.

**Weaknesses:**

- I think that the claim that Mamba "just does MLE" is an overstatement. That is, the argument that Mamba = SSIB with $\beta = 0$ is  true only for an idealized training loss, without regularization. In practice, large-scale Mamba training uses regularization (dropout, weight decay, data noise, possibly distillation) that already limits I(H;X).

- For me, the Bernoulli parameterization of $A_t$ is not fully convincing. The paper says "it is natural to model $A_t$ as Bernoulli" because each entry decides keep/drop. But in Mamba-2 the reason $A_t \in (0,1)^{N \times D}$ is the specific discretization, which leads to entries being viewed as *decay factors*, not probabilities. Turning *every* entry into an i.i.d. Bernoulli with parameter $p_t$ is very a strong modeling assumption (e.g., it allows exactly zeros which implies full forgetting about the context).

- For Bernoulli the paper says that Gumbel–Softmax (Binary Concrete) is used, which is fine. For the Exponential variant there is no information about the differentiability of the estimator.

- The "importance" score simply averages $A_t$ over both the state dimension ($N$) and the feature dimension ($D$). That is exactly the kind of averaging MambaAttention/MambaLRP later criticized as washing out channel-specific effects.

- For interpretability, the evaluation would be stronger if the paper includes other SSM-specific methods, such as MambaAttention [1], MambaLRP [2], and LATIM [3]. The current baselines are somewhat weak and outdated.

- On 370M / 20B-token pretraining, the improvements over Mamba-2 are small and vary across tasks. On datasets like MMLU and WinoGrande at that scale the noise is high, so it is hard to attribute the gains to the IB formulation. The 7B finetuning comparison is also slightly unfair because Baleen is finetuned on the extra 0.5B Crystal data but the baseline Mamba-2 is not.

[1] https://arxiv.org/abs/2403.01590
[2] https://arxiv.org/abs/2406.07592
[3] https://aclanthology.org/2025.acl-long.1194/

**Questions:**

- What is the Crystal dataset used for 7B instruction tuning? I could not find further information about it.

- For the Exponential case, what is the exact gradient estimator used?

---

> ### Author Response · Authors · 2025-11-28
>
> Thank the reviewer for the insightful feedback. Responses are as below.
>
> > I think that the claim that Mamba "just does MLE" is an overstatement. That is, the argument that Mamba = SSIB with is true only for an idealized training loss, without regularization. In practice, large-scale Mamba training uses regularization (dropout, weight decay, data noise, possibly distillation) that already limits I(H;X).
>
> This argument mainly hinges on the distinction between IB and standard regularizers. L2 penalties or dropout act directly on **parameters**, with essentially the same implementation across architectures (including Baleen). In contrast, IB explicitly compresses **representations**. While parameter regularizers can implicitly reduce the complexity of the representation space $\mathcal{H}$, there is no guarantee that the resulting representations H are compressed with respect to the information in X. Therefore, the two types of regularization are orthogonal, and β is still 0 for pure Mamba since **parameters regularizer can only affect $\mathcal{H}$ but not H** .
>
> > For me, the Bernoulli parameterization of is not fully convincing...
>
> We agree this is a too strong modeling assumption in principle, but in practice we never use hard Bernoulli sampling: we implement via a Gumbel–Softmax relaxation, so each gate still behaves as a continuous decay factor, with an additional force that sharpens the separation between “keep” and “drop” cases.
>
> > For Bernoulli the paper says that Gumbel–Softmax (Binary Concrete) is used, which is fine. For the Exponential variant there is no information about the differentiability of the estimator.
>
> Please refer to response to Q2.
>
> > The "importance" score $A\_t$ simply averages
>  over both the state dimension ($D$) and the feature dimension ($N$). That is exactly the kind of averaging MambaAttention/MambaLRP later criticized as washing out channel-specific effects.
>
>
>
> Yes. Compared to MambaAttention, we do not need to **additionally** compute separate attention scores; compared to MambaLRP, our model is inherently/self-interpretable. Our results show that, despite the simple averaging, the resulting importance scores are both effective and faithful.
>
> > For interpretability, the evaluation would be stronger if the paper includes other SSM-specific methods, such as MambaAttention [1], MambaLRP [2], and LATIM [3]. The current baselines are somewhat weak and outdated.
>
> Thanks for you valuable suggestion. We will add MambaAttention [1], MambaLRP [2] in our next revision.
>
> > On 370M / 20B-token pretraining, the improvements over Mamba-2 are small and vary across tasks. On datasets like MMLU and WinoGrande at that scale the noise is high, so it is hard to attribute the gains to the IB formulation. The 7B finetuning comparison is also slightly unfair because Baleen is finetuned on the extra 0.5B Crystal data but the baseline Mamba-2 is not.
>
> Regarding the 370M “small improvement”: we would like to emphasize a **long-standing issue in trustworthy ML, namely the trade-offs between accuracy–interpretability and accuracy–robustness** [1,2]. It is very hard to improve all three simultaneously, and accuracy is often partially sacrificed to gain better interpretability and robustness (no free lunch). Our method sets a new Pareto front by improving robustness and interpretability while still matching or slightly improving accuracy.
>
> Regarding 370M evaluation noise: high variance can change the exact numbers but typically does not change the ranking. In our experiments, the ranking of baseline models is consistent with the results reported for Gated-DeltaNet under different training recipes.
>
> Regarding the 7B evaluation: we apologize for the confusion. We do finetune the Mamba-2 baseline on the same additional 0.5B Crystal data, so the comparison is fair.
>
> ---
> > Q1. What is the Crystal dataset used for 7B instruction tuning? I could not find further information about it.
>
> The Crystal data is used to train CrystalCoder and CrystalChat in LLM360 [3]. It is a mix of publicly available datasets, as described in the Hugging Face card for LLM360/CrystalChat. The dataset itself is not released yet and we obtained a copy directly from the authors.
>
> > Q2. For the Exponential case, what is the exact gradient estimator used?
>
> No estimator is needed, and we apologize for the confusion. For Baleen-B, we use the Gumbel–Softmax reparameterization, which makes the discrete sampling operation differentiable. For Baleen-E, the exponential gates are continuous and already differentiable, so gradients are computed via standard backpropagation.
>
> ---
> **References**:
>
> [1] Zhang et al., Theoretically Principled Trade-off between Robustness and Accuracy, ICML 2019.
>
> [2] Rudin, Stop explaining black box machine learning models for high stakes decisions and use interpretable models instead, Nat. Mach. Intell. 2019.
>
> [3] LLM360 Team, LLM360: Towards Fully Transparent Open-Source LLMs, 2024.

---

### Official Review · Reviewer_pniM · 2025-11-03

**Soundness:** 3
**Presentation:** 3
**Contribution:** 2
**Rating:** 4
**Confidence:** 3

**Summary:**

This paper introduces Baleen, a new family of SSMs that integrates stochastic selective memory with an information bottleneck objective to improve interpretability and robustness. Extending Mamba, Baleen models state-selection gates as random variables with a variational KL regularizer, balancing context retention and compression while preserving linear-time recurrence. Two variants based on Bernoulli gating and exponential time sampling, Baleen-B and Baleen-E, yield self-interpretable token-level importance scores derived from expected gate values. The formulation provides an information-theoretic refinement of Mamba, explicitly controlling how much input history is retained, and promoting minimal sufficient representations that enhance robustness and explanation fidelity.

**Strengths:**

1. The paper presents a principled integration of information bottleneck theory into SSMs, yielding stochastic gate regularization that improves both interpretability and robustness.

2. The proposed Bernoulli and exponential gating variants are theoretically well-motivated and retain Mamba’s linear computational complexity.

3. The method demonstrates strong robustness under input perturbation and adversarial settings.

**Weaknesses:**

1. While formulating a variational information bottleneck on Mamba’s gating is interesting, the conceptual novelty is incremental. The method essentially adds a KL-based sparsity regularizer to encourage selective gating, similar to variational dropout or selective masking. The core SSM architecture remains unchanged, and the stochastic gates are parameterized similarly to Mamba’s deterministic ones. Thus, Baleen may be viewed as “Mamba2 + KL regularizer,” raising the question of whether similar gains could be achieved with simpler methods (e.g., L0​/L1​ penalties or tuned dropout). Nonetheless, the formalization through the IB framework provides a theoretically grounded lens on sparsity and interpretability, which may justify the contribution despite its incremental nature.

2. Baleen inherits the fixed hidden-state constraint of SSMs, which limits the amount of information it can store and retrieve. This becomes problematic for tasks requiring the distributed recall of multiple facts across long contexts. The IB objective improves information quality but may over-compress, discarding useful signals needed later. Since the number of independent bits is bounded by the state dimension, Baleen cannot fully overcome memory saturation when the sequence length greatly exceeds the state size. Without adaptive or external memory, it may still struggle with tasks requiring long-term or multi-fact reasoning.

3. Although Baleen consistently outperforms prior SSMs, its empirical gains are modest. Improvements vary across tasks, with some regressions, and certain baselines (e.g., Gated-DeltaNet) outperform it on individual benchmarks. Moreover, comparisons are limited to SSM and linear RNN families; Transformer baselines, or hybrid models of comparable scale, are missing. Without such results, it remains unclear whether Baleen approaches or still lags Transformer-level performance.

**Questions:**

1. How sensitive are Baleen’s results to the choice of the IB regularization strength (the weight β in KL term in Eq. 4)?

2. Can Baleen effectively handle tasks requiring many pieces of information to be retained over long sequences?

3. Since the gates are stochastic, have the authors examined whether the expected gate values are stable across different runs, and whether the top-selected tokens align with known important words or ground-truth rationales?

---

> ### Author Response · Authors · 2025-11-28
>
> Thank the reviewer for the insightful feedback. Responses are as below.
>
> > Weakness 1: While formulating a variational information bottleneck on Mamba’s gating is interesting, the conceptual novelty is incremental...Thus, Baleen may be viewed as “Mamba2 + KL regularizer,” raising the question of whether similar gains could be achieved with simpler methods (e.g., L0/L1 penalties or tuned dropout).
>
> This concern mainly hinges on the distinction between IB and standard regularizers. L0/L1 penalties or tuned dropout act directly on **parameters** and are model-agnostic, with same implementations across models. In contrast, IB explicitly compress **representation**, and instantiated differently for each architecture. Therefore, the two kinds of regularizers are orthogonal and parameter regularizers are often already tuned like using e-4 weight decay (L2 penalty) and 0 attention drop out.
>
> Our contribution is to extend the IB principle to SSM gating but not incremental. This requires a **model-specific formulation and implementation** (where to place the bottleneck, how to parameterize and train the variational gates) rather than simply adding a generic KL term. The backbone SSM remains Mamba-like is just as in most IB work that keeps the base CNN/Transformer.
>
> > Weakness 2: ...Without adaptive or external memory, it may still struggle with tasks requiring long-term or multi-fact reasoning.
>
> Please refer to response to Q2.
>
> > Weakness 3: Although Baleen consistently outperforms prior SSMs, its empirical gains are modest and certain baselines (e.g., Gated-DeltaNet) outperform it on individual benchmarks... Moreover, comparisons are limited to SSM and linear RNN families; Transformer baselines, or hybrid models of comparable scale, are missing...
>
> Regarding empirical gains: “Small” margins are typically meaningful for LLM. For instance, a **≈0.5-point improvement over Mamba2** is what made Gated DeltaNet the choice in several large-scale industrial systems (e.g., Qwen3-Next, Kimi’s linear backbone). In our experiments, Baleen improves **over Gated DeltaNet and by roughly 1 point over Mamba2**, while simultaneously enhancing robustness and interpretability. Besides, the occasional drop on individual benchmarks are inherent to the **well-documented robustness–accuracy  and interpretability–accuracy trade-offs** [1,2] in trustworthy ML. But ours pushes the Pareto front towards better robustness/interpretability/accuracy.
>
> Regarding comparison to Transformers: Modern LLaMA-style Transformers are stronger than purely linear models, which is expected given their higher compute (roughly >1.5× more for the same sequence length [3]). But Gated DeltaNet (GDN) already demonstrates that hybrid GDN–attention models can outperform strong Transformer baselines, and the rankings of linear models are strongly correlated their hybrid counterpart; **since Baleen outperforms both Mamba2 and GDN in the pure SSM regime, we expect hybrid “Baleen + attention” variants to be stronger than these Transformer and hybrid baselines.**
>
> ---
>
> > Q1. How sensitive are Baleen’s results to the choice of the IB regularization strength (the weight β in the KL term in Eq. 4)?
>
> We find Baleen is not very sensitive to β: we simply fix β = 0.1 for Baleen-B and β = 1e−5 for Baleen-E without hyper-parameter tuning. These values are chosen so that the KL term has a similar magnitude to the cross-entropy loss. We will add ablation results in the appendix to further support this insensitivity.
>
>
>
> > Q2. Can Baleen effectively handle tasks requiring many pieces of information to be retained over long sequences?
>
> Our robustness experiments show that Baleen suppresses over-attention to very recent tokens, which suggests it uses its fixed state to attend to longer context token than vanilla Mamba and should therefore handle long sequences better in practice. Although long-context performance is not the main focus of this work (we focus on robustness and interpretability without sacrificing accuracy), we will add a needle-in-a-haystack evaluation to empirically support this claim.
>
>
> > Q3. Since the gates are stochastic, have the authors examined whether the expected gate values are stable across different runs, and whether the top-selected tokens align with known important words or ground-truth rationales?
>
> During training the gates are stochastic, but at inference we directly use their expectations, as stated in line 283 of the paper. This makes the predictions and top-selected tokens deterministic for a fixed input and generation configuration, and thus stable across repeated runs, which is also observed in experiments.
>
> ---
> **References:**
>
> [1] Zhang et al., Theoretically Principled Trade-off between Robustness and Accuracy, ICML 2019.
>
> [2] Rudin, Stop explaining black box machine learning models for high stakes decisions and use interpretable models instead, Nat. Mach. Intell. 2019.
>
> [3] Wang et al., Linformer: Self-Attention with Linear Complexity, 2020.

---

### Meta-Review · Area_Chair_woXJ · 2025-12-21

**Summary:**

This paper proposes Baleen, a family of stochastic state space models that extend Mamba by introducing an information bottleneck (IB) objective through variational gating. Across reviewers, there is consensus that the paper is well written, technically sound, and explores an interesting direction—namely improving robustness and interpretability of SSMs via selective memory. However, reviewers consistently raise concerns that the conceptual novelty is incremental, the empirical gains are modest and sometimes noisy, and several key claims are insufficiently validated within the current submission. While the rebuttal clarifies motivations and implementation details, most substantive concerns are deferred to future revisions rather than being resolved during the rebuttal phase.

**Reviewer Concerns:**

**Concerns that were partially addressed:**

The authors clearly explain the distinction between information bottleneck regularization and parameter-level regularization (e.g., L1/L2, dropout), and clarify inference-time behavior (expectation of stochastic gates).

Questions regarding gradient estimation, hyperparameter sensitivity, and fairness of 7B finetuning comparisons are adequately clarified.

Connections to prior work (e.g., Stuffed Mamba) are acknowledged and conceptually aligned.

**Concerns that remain outstanding:**

Multiple reviewers view Baleen as a principled but relatively small extension of existing Mamba-style gating, and the IB framing, while elegant, does not clearly translate into a qualitatively new modeling capability.

 Improvements over strong SSM baselines are often small, vary across benchmarks, and lack statistical robustness analysis. Promised ablations, regime studies (sequence length × state size), and additional evaluations are not present in the current version.

Transformer or hybrid baselines are not included, and the argument that linear-model rankings transfer to hybrid settings remains speculative.

The proposed importance scores rely on heavy averaging, and stronger SSM-specific interpretability baselines (e.g., MambaAttention, MambaLRP) are only promised for future revisions.

Motivation and evidence gap: Claims about Mamba’s failure to forget and its connection to robustness are largely intuitive and not sufficiently supported by direct empirical analysis in this paper.

***Overall, several reviewers note that the rebuttal contains many “we will add” responses, indicating that the work is not yet fully mature at the time of evaluation.***

**Reviewer Scores:**

Reviewer pniM: Likely unchanged. Rebuttal clarifies intent but does not resolve concerns about novelty and memory limits.

Reviewer BgmA: Likely unchanged. Clarifications help, but interpretability baselines and empirical strength remain insufficient.

Reviewer vu9z: Likely unchanged. Motivation and regime analysis are still largely deferred.

Reviewer yxmK: Likely unchanged or slightly lower confidence in acceptance, given low confidence and acknowledgment that rejection would be reasonable.

---

### Decision · Program_Chairs · 2026-01-26

Reject